# Fibrous Roots of Cimicifuga Are at Risk of Hepatotoxicity

**DOI:** 10.3390/molecules27030938

**Published:** 2022-01-29

**Authors:** Yang Yu, Jialiang Tan, Jianing Nie, Chongning Lv, Jincai Lu

**Affiliations:** 1School of Traditional Chinese Materia Medica, Shenyang Pharmaceutical University, Shenyang 110006, China; yuyang_94@outlook.com (Y.Y.); tanjl20706@outlook.com (J.T.); njn521521@outlook.com (J.N.); 2Liaoning Provincial Key Laboratory of TCM Resources Conservation and Development, Shenyang Pharmaceutical University, Shenyang 110006, China

**Keywords:** hepatotoxicity, fibrous roots, rhizome, *Cimicifuga dahurica* (Turcz.) Maxim

## Abstract

The cause of liver damage by using black cohosh preparation has been concerned but remains unclear. After a preliminary investigation, the black cohosh medicinal materials sold in the market were adulterated with Asian cohosh (Cimicifuga) without removing the fibrous roots. The safety of Cimicifuga rhizome and fibrous roots is unknown and has not been reported. Therefore, in this paper, the rhizome and fibrous roots of *Cimicifuga dahurica* (Turcz.) Maxim (*C. dahurica*) were completely separated, extracted with 70% ethanol, and freeze-dried to obtain crude rhizome extract (RC) and fibrous roots extract (FRC). UHPLC-Q-TOF-MS was used to identify 39 compounds in the rhizome and fibrous roots of Cimicifuga, mainly saponins and phenolic acids. In the L-02 cytotoxicity experiment, the IC_50_ of fibrous roots (1.26 mg/mL) was slightly lower than that of rhizomes (1.417 mg/mL). In the 90-day sub-chronic toxicity study, the FRC group significantly increased the level of white blood cells, ALP, ALT, AST, BILI and CHOL (*p* < 0.05); large area of granular degeneration and balloon degeneration occurred in liver tissue; and the expression of p-NF-kB in the nucleus increased in a dose-dependent manner. Overall, Fibrous roots of Cimicifuga are at risk of hepatotoxicity and should be strictly controlled and removed during the processing.

## 1. Introduction

Black cohosh extract can produce estrogen-like effects and regulate endocrine balance, so it is widely used as a hormone replacement and an anti-inflammatory agent for the treatment of menopausal syndrome. Since around 2000, clinical cases of liver injury caused by taking black cohosh extract appeared successively in Australia, the United States and European countries [1,2,3,4]. However, according to the current study, the cause of liver injury caused by black cohosh preparation has not been found [5,6,7,8]. With the shortage of black cohosh resources, some herbs similar to black cohosh have been adulterated into the medicinal material market. Many Asian species of cohosh (*Cimicifuga foetida*, *C. dahurica* and *C. heracleifolia*) are sold as black cohosh [9].

The Asian species of cohosh-cimicifuga has a long history of medicinal use in Asian countries such as China, Japan and Korea [10]. The main chemical constituents of Cimicifuga are triterpenoid glycosides, Phenylpropanes, nitrogenous compounds and chromones [11]. Among them, saponins have the functions of regulating immunity, protecting the brain and heart, delaying aging, antitumor, expanding cerebral vessels and increasing blood flow [12,13]. Phenylpropanoids have antitumor, anti-HIV, antioxidant, anti-inflammatory, anti-microbial, anticoagulant and other biological activities. Some phenylpropanoids also have the effects of lowering blood lipid, blood pressure, blood glucose, antithrombotic, antimutagenic, analgesia and sedation [14]. Alkaloids have anti-inflammatory, antibacterial, vasodilator, cardiotonic, anti-asthmatic, anticancer and other effects [15]. Similar to black cohosh, it is also used to relieve menopausal symptoms [16].

Cimicifuga has been used in China for thousands of years, and no clinical cases of liver injury have been reported so far. We found that the processing method of Cimicifuga in the Chinese Pharmacopoeia clearly stipulates “removal of soil, sun-drying, singe or remove the fibrous roots”. While in the west, there was no such operation in the preparation of black cohosh. In addition, our previous studies found that the chemical composition and content of Cimicifuga rhizome and fibrous roots were different, and cimicifugone A and B isolated from fibrous roots of cimicifuga may have hepatotoxicity.

Therefore, it is necessary to comprehensively evaluate the toxicity of the rhizome and fibrous roots of cimicifuga. In this study, UHPLC-Q-TOF-MS was used to identify the main chemical components, and the hepatotoxicity was evaluated by the cytotoxicity test on normal human hepatocytes (L-02) in vitro and the 90-day sub-chronic toxicity test in vivo.

## 2. Results and Discussion

### 2.1. Identification of the Chemical Compositions by UHPLC-Q-TOF-MS

In order to identify the main chemical components of cohosh, firstly, a comprehensive search and classification of the compounds of cimicifuga species was carried out in the Scifinder database, and a potential database was initially established. Then, by comparing the retention time and the accurate relative molecular mass, the structure of the detected chemical composition is determined. In addition, we combined our laboratory’s previous experience in the isolation and identification of cimicifuga and summarized the fragmentation regularities of these compounds by mass spectrometry [10,17,18]; some compounds were determined by comparison with the mass spectrometry data of the reference substance. Figure 1 shows the UHPLC-Q-TOF-MS base peak intensity chromatograms of the rhizomes and fibrous roots of *C. dahurica* in positive and negative electrospray (ESI) modes. Finally, 52 compounds in the rhizome extract (RC) and fibrous roots extract (FRC) of *C. dahurica* were analyzed. The corresponding mass spectra data are listed in Table 1 and Table 2. The analysis results showed that the main chemical components of the rhizomes and fibrous roots of *C. dahurica* are all saponin and phenolic acid compounds, in addition to a small number of alkaloids.

### 2.2. Cytotoxicity Results of L-02 Cells

In the in vitro cytotoxicity test, the MTT method was used to detect the effects of the RC and FRC of *C. dahurica* on the viability of normal human liver cells L-02. RC and FRC have no obvious inhibitory effect on the proliferation of L-02 cells, but it is worth noting that the IC_50_ of the FRC (1260 μg/mL) is slightly lower than the IC_50_ value of the RC (1417 μg/mL), suggesting that the fibrous roots may have a higher potential for hepatotoxicity than the rhizome.

### 2.3. General Observation and Body Weight

During the whole experiment period, the weight of female and male rats changed normally, and there was no significant difference between the treatment groups and the control group (Figure 2). The diet and water intake of each group were normal. However, it is worth noting that a total of six rats died during the experiment, of which the mortality rate of the fibrous roots group and rhizome group was 5:1, while the death ratio of female and male rats was 2:1. These results suggested that the toxicity risk of fibrous roots was significantly higher than rhizome, and the females were more easily damaged than males.

### 2.4. Urinalysis, Hematology and Biochemical Analysis

The results of urinalysis are shown in Table 3. No dose-related adverse reactions were observed in any administration group of female and male rats. There are some slight gaps in various test indicators, but these changes are sporadic and within the acceptable range of normal physiology.

The results of the hematology test are shown in Table 4. For female rats, compared with the control group, the white blood cell (WBC) level of each dose group of FRC was significantly increased (*p* < 0.05), but there was no significant difference in the RC group. For male rats, only the FRC-UHD group had a significant increase in WBC (*p* < 0.01). As the “guard” of the human body’s fight against diseases, WBC can deform and pass through the capillary wall to concentrate on the invasion site of the disease, enveloping and swallowing the disease [19,20]. The number of WBC in the body is higher than the normal value, so it is likely that the body has inflammation. Hematological test results show that long-term use of fibrous roots is likely to cause inflammation, and females are more sensitive than males.

The results of serum biochemical analysis are shown in Table 5. Compared with the control group, there was no significant difference in the biochemical indicators of the RC groups (*p* > 0.05). Yun [21] selected the crude extract of the rhizome of *Cimicifuga heracleifolia* Kom. for 13 weeks of safety evaluation, and the biochemical test results showed that the changes in various indicators are sporadic and within physiologically acceptable ranges. This is consistent with the results of the rhizome of *C. dahurica* in this study, suggesting that the rhizome part of Cimicifuga is safe and will not cause physical lesions under normal clinical doses.

The FRC groups had a significant increase in Alanine aminotransferase (ALT) and aspartate aminotransferase (AST) in female rats (*p* < 0.05), and the content of ALT increased significantly (*p* < 0.001) in the FRC-UHD group. ALT and AST are important clinical indicators of liver function tests to determine whether the liver is damaged [22,23]. It can be seen from the test results that long-term use of FRC extract may cause liver damage, thereby increasing the content of AST in the blood. At the same time, it may be accompanied by a biliary obstruction, leading to ALT excretion obstacles and backflow into the blood, making the serum ALT content significantly increased. At the same time, for male rats, the FRC-UHD group also showed a significant increase in Alkaline phosphatase (ALP) and ALT levels (*p* < 0.001). ALP is excreted through the hepatobiliary system. When the liver is damaged or impaired, liver cells over-produce ALP and enter the blood through the lymphatic tract and sinusoids, causing high levels of ALP in serum [24,25].

Most of the body’s bilirubin (TBIL) comes from hemoglobin released by the lysis of senescent red blood cells, including indirect and direct bilirubin. Total gallbladder = direct bilirubin + indirect bilirubin. Indirect bilirubin is transported to the liver through the blood, and direct bilirubin is produced through the action of hepatocytes [26]. The total bilirubin content in the blood of male rats after administration of fibrous roots was generally higher than that in the control group, and the FRC-UHD group was statistically significant (*p* < 0.05). It is suggested that the liver of rats after taking fibrous roots is damaged, and the process of direct bilirubin transport through the blood to the liver will be correspondingly inhibited, thereby increasing the total bilirubin content in the blood. Total cholesterol (CHOL) includes free cholesterol and cholesterol esters. The liver is the main organ for cholesterol synthesis and storage, and its serum concentration can be used as an indicator of lipid metabolism [27]. The total cholesterol content of the FRC-UHD group was significantly higher than that of the blank group, and the female rats were significantly (*p* < 0.01) higher than that of the male rats (*p* < 0.05). It is suggested that excessively high doses of fibrous roots are likely to cause damage to the liver of rats, leading to obstruction of bile excretion, increasing the synthesis of blood lipoproteins and cholesterol in the liver, and increasing free cholesterol, leading to high total cholesterol.

In addition, for male rats, the urea nitrogen (UREA) and creatinine (CREA) of the fibrous roots administration group increased to varying degrees but did not show a dose dependence. When the kidney’s ability to eliminate and excrete decreases, it will also lead to an increase in blood urea nitrogen, which is often accompanied by an increase in blood creatinine [28]. This suggests that the fibrous roots of *C. dahurica* may also cause slight damage to the kidneys.

### 2.5. Organ Weights and Histopathological Changes

The organ weights and organ coefficients are shown in Table 6. By comparing the absolute and relative organ weights of the main organs of rats in each dose group of RC and FRC, it was found that there was no significant difference compared with the control group (*p* > 0.05). Visual observation of the organs showed that the RC group and the FRC group were similar in volume, color and shape to the control group, and there was no obvious damage or pathology.

Histopathological examination of the liver (Figure 3) found that the liver tissue sections of the female and male rats in the control group were basically normal. The liver tissue envelope was composed of dense connective tissue rich in elastic fibers. The liver lobules were clearly demarcated and arranged regularly. The central part of the hepatic lobule is the central vein, surrounded by roughly radially arranged liver cells and hepatic sinusoids. The liver cells are round and full, the liver plates are arranged regularly and neatly and the sinusoids are not significantly expanded or squeezed. The portal area between adjacent liver lobules has no obvious abnormalities and no obvious inflammatory changes. Compared with the control group, the liver tissue sections of each dose of the RC administration group showed no significant difference, and this is consistent with the research report of Mazzanti [7]. Mazzanti conducted morphological tests on liver samples after oral administration of black cohosh extract to Wistar rats for 30 days, and the results showed that black cohosh extract had no significant effect on the liver morphology of rats. However, in the liver tissues of the FRC-MD and FRC-HD groups, a large number of granular degeneration of hepatocytes, swelling of the cells, loose cytoplasm and a fine granular shape were seen (yellow arrows). In the FRC-UHD group, a large number of balloon-like degeneration of hepatocytes, swelling of the cells, centered nucleus and vacuolation of the cytoplasm were seen in the tissues of the FRC-UHD group (Black arrows). It is suggested that fibrous roots of *C. dahurica* caused liver tissue lesions.

### 2.6. Effect on Protein Expression of p-NF-κB

This study explored the molecular mechanism of drug-induced liver injury by detecting the expression of p-NF-κB in the nucleus of rat liver tissue. Western blot results are shown in Figure 4. In the control group, p-NF-κB was almost not expressed. Compared with the control group, there was no statistical difference in the expression of p-NF-κB in the RC group at the dose of this experiment (*p* > 0.05), but there was still a small amount of protein expression in the liver cells of female rats, suggesting that long time exposure or an increased dose may lead to increased risk of hepatotoxicity to female. In accordance with the above experimental results, the expression of p-NF-κB in the hepatic nuclei of both male and female rats was significantly increased in the FRC group (*p* < 0.01) in a dose-dependent manner. In addition, the expression of p-NF -κB in female rats was significantly higher than that in male rats. NF-κB is a heterodimer combined with inactivated inhibitory protein (IκB). It is located in the cytoplasm and is an important transcription factor in the process of inflammation. Long-term administration of cohosh root causes liver damage and inflammation in rats, which activates the phosphorylation of NF-κB and transfers to the nucleus, increasing the expression of p-NF -κB in the nucleus of rats. The p-NF -κB activates specific target genes through transcription, e.g., inflammatory mediators such as cytokines, chemokines, cell adhesion molecules, etc. [29,30]. This translocation is the mechanism by which cells respond to oxidants or inflammatory and immune stimuli. It suggests that the fibrous roots of *C. dahurica* can cause liver tissue damage in rats and trigger inflammation.

## 3. Materials and Methods

### 3.1. Plant Material and Animals

*C**.dahurica* was purchased from the Shenyang medicine market in China and authenticated by Prof. Jin-cai Lu (School of Traditional Chinese Materia Medica, Shenyang Pharmaceutical University). The medicinal material specimens are stored in the Department of Traditional Chinese Medicine Resources of Shenyang Pharmaceutical University (No. 2019102301). We artificially separate the rhizomes and fibrous roots when *C. dahurica* is fresh and cut them into small pieces after sun-drying. Then, they were extracted with 70% ethanol at a factory, concentrated and freeze-dried. (The extraction yield of the alcohol extraction of RC was 0.32 g/g, and the extraction yield of the alcohol extraction of FRC was 0.29 g/g).

### 3.2. Identification of Chemical Compositions in Crude Extract

Using Compact™ QTOF mass spectrometry (Bruker Daltonik GmbH, Bremen, Germany) and elution autosampler UHPLC system tandem technology (Bruker Daltonik GmbH, Bremen, Germany), the crude extracts of rhizomes and fibrous roots of *C. dahurica* were qualitatively analyzed in positive and negative electrospray (ESI) mode at *m*/*z* 50–1300. A Waters Acquity BEH C18 column (2.1 × 100 mm, 1.7 μm) (Waters Corp., Milford, MA, USA) was used. The mobile phase consisted of water-formic acid (100:0.1, *v*/*v*) (A) and acetonitrile (B). The injection volume was 1 μL, and the flow rate was 0.3 mL/min. The column temperature was maintained at 35 °C. The optimized gradient elution conditions for UHPLC are: 15~25% B in 0~10 min, 25~40% B in 10~15 min, 40~47% B in 15~25 min and 25~26 min within 47~15% B. The instrument parameters used for mass spectrometry are as follows: At 1.8 Bar, 220 °C and a flow rate of 8.0 L/min, and nitrogen is used as the nebulizer gas. Collision energy values: 100 *m*/*z*, 20 eV; 500 *m*/*z*, 30 eV; 1000 *m*/*z*, 50 eV; and *m*/*z* 1300, 55 eV. All data were collected and processed using Bruker Compass data analysis 5.1 software (Bruker Daltonik GmbH, Bremen, Germany).

### 3.3. In Vitro Hepatocyte Toxicity Test

After resuscitation, human normal hepatocytes L-02 were cultured in a DMEM medium containing 10% fetal bovine serum and 1% double antibody. As for the cells, they were cultured in an incubator at 37 °C and 5% CO_2_ at saturated humidity. The cell density grew. When it reaches 70–80%, trypsin was added to digest and pass it down to a new cell culture dish. The L-02 cells in the logarithmic growth phase were seeded in a 96-well plate at 5 × 103 cells/well, with 100 μL of cell culture medium per well. The 96-well plate was placed in an incubator for about 12 h to adhere to the cell wall and then starved for another 12 h. Then, the serum-free culture medium replaced with different concentrations of drugs (100, 200, 400, 800, 1600 μg/mL). For the medium of drug RC or FRC and 2% serum, the control group was replaced with an equal volume of 2% serum-containing culture medium and returned to the incubator to continue culturing for 72 h. After the culture plate was removed, we added μL of MTT working solution to each well and incubated it in the incubator for 4 h. Then, all the liquid in the wells was aspirated; 150 μL of DMSO was added to each well to dissolve the crystals; the culture plate was placed on a shaker and shook for 10 min to fully dissolve the crystals. We used a multifunctional microplate reader to detect the absorbance (OD) value at a 492 nm wavelength. The experiment was repeated three times [30]. Finally, we calculated the cell survival rate according to the following formula and used SPSS 19.0 software to calculate the IC_50_ value of the drug. Calculated as follows:

Inhibition rate of cell proliferation (%) = (1 − OD administration group/OD control group) × 100%.

### 3.4. Experimental Design for the Oral Toxicity Study

One hundred and eight Special pathogen-free (SPF) Sprague Dawley rats (aged 4~8 weeks, weight 165 ± 15 g) were provided by the Experimental Animal Center of Shenyang Pharmaceutical University (Shenyang, China), and they were used after a week of quarantine and acclimatization. Animals were housed in an SPF laboratory under controlled temperatures (25 ± 1 °C), humidity (60 ± 10%) conditions and a 12 h light/dark cycle, and they were allowed free access to a balanced murine diet and water during the experiment (Lab Diet 5002 Certified Rodent Diet). All of the animal experiments were approved by the Institutional Animal Care and Use Committee of the Biomedical Research Institute at the Shenyang Pharmaceutical University and complied with the National Institutes of Health guide for the care and use of Laboratory animals (NIH Publications No. 8023, revised 1978). Combined with the previous acute toxicity experiment and the clinical dosage of cimicifuga, the dosage for the 90-day repeated administration toxicity experiment was determined. The sample of RC and FRC was Orally administered (10 mL/kg) to rats (6/sex/group) at doses of 0.585 g/kg (LD), 1.755 g/kg (MD), 5.265 g/kg (HD) and 15.795 g/kg (UHD), and the normal control group was given vehicle (0.5% CMC-Na solution) once a day. During the experiment, we observed the test animals’ general clinical symptoms, food and water consumption and mortality every day. Additionally, we measured their body weight once a week.

### 3.5. Urinalysis, Hematology and Biochemistry Analysis

In the last week of treatment, fresh urine samples of 10 rats (5 males and 5 females) were taken from each group and analyzed with a urine analyzer (URIT-500B, Guangxi, China) to evaluate the following parameters: WBC, KET, NIT, URO, BIL, PRO, GLU, SG, BLD, pH and Vc.

The rats were anesthetized with isoflurane after the final gavage. Blood samples are collected through the posterior vena cava. Collect the whole blood sample into the EDTA blood collection tube. Use the automatic hematology analyzer ADVIA 2120i hematology counter (Siemens, Germany) to perform hematology determination of the following parameters: WBC, RBC, HGB, HCT, PLT, MCV, MCH, MCHC, neutrophils, eosinophils, basophils, lymphocytes, monocytes and reticulocytes.

For the biochemical analysis, whole blood was separated at 3000 rpm for 15 min, and serum was separated immediately. Measured with automatic biochemical analyzer 7020 (Hitachi, Tokyo, Japan): ALP, ALT, AST, BUN, CREA, TBIL, GLU, TC, TG, TP, ALB.

### 3.6. Gross Findings, Organ Weights, and Histopathological Assessments

During necropsy, the organs and tissues were observed macroscopically, and the absolute weight of the heart, liver, spleen, lung, kidney, stomach, brain, adrenal gland, thymus and genitalia were recorded, and the relative weight (organ-to-body weight ratios) was calculated.

Part of the liver tissue (2.0 cm × 2.0 cm × 0.3 cm) was taken and fixed in 4% paraformaldehyde solution, embedded in paraffin, sectioned, and stained with hematoxylin and eosin. Then, use an optical microscope for pathological examination to analyze whether there are histological changes in the liver of each group of rats.

### 3.7. Western Blot Analysis

We collected 60mg liver samples of rats in each experimental group and extracted nuclear protein according to the instructions of the nuclear protein extraction kit. The protein concentration was determined by the bicinchoninic acid (BCA) method. Equal protein samples were separated using sodium dodecyl sulfate polyacrylamide gel electrophoresis (Bio-Rad, Hercules, CA, USA) and electro transferred onto polyvinylidene difluoride (PVDF) membranes for 2 h. The membranes were incubated for 1 h with 5% skim milk in the Tris-buffered saline containing Tween 20 to block nonspecific binding. After incubating at room temperature for 1 h, the blocking solution was discarded, and the primary antibodies of Phospho-NF-κB p65 (3033, 1:1000) and β-actin (4970, 1:1000) (Cell Signaling Technology, Danvers, MA, USA) were added to the membranes, which were incubated at 4 °C overnight and then washed with Tris buffered saline (TBS) containing 0.1% Tween-20 (TBST) buffer for 3 × 10 min. The secondary antibodies Anti-rabbit IgG (7074, 1:1000) (Cell Signaling Technology, Danvers, MA, USA) were added and incubated for 2 h at room temperature, after washing three times with TBST, and developed with enhanced chemiluminescence (ECL) developer and exposed to X-ray film. The expression levels of β-actin were used as control.

### 3.8. Statistical Analysis

All of the values are expressed as mean ± SD. The statistical analysis was performed using a one-way ANOVA, followed by a multiple comparison procedure with a Tukey/Duncan test using SPSS software version 19.0 (IBM corporation, Armonk, NY, USA). *p* values of less than 0.05 were considered to be statistically significant. The figures were conducted using GraphPad Prism 7.0 (GraphPad Software, La Jolla, CA, USA).

## 4. Conclusions

This study found that the rhizome of *C. dahurica* was safe to take for a long time under the dose of 15.795 g/kg crude drug a day in rats. (It is equivalent to 177 g/d of clinical intake for adults with a weight of about 70 kg.) Additionally, there were no obvious toxic and side effects. However, there is a risk of liver damage in the long-term high-dose administration of *C. dahurica* fibrous roots since with an increase in the dose, the number of white blood cells in the blood increases significantly, and the biochemical test shows ALP, ALT, AST, BILI and CHOL all have a significant increase, and the expression of p-NF-κB is also significantly increased, which are all landmark indicators of liver damage. This result explains the rationality and necessity of removing fibrous roots in the processing method of cimicifuga. In addition, the material basis of hepatotoxicity of Cimicifuga fibrous roots remains to be further studied.

## Figures and Tables

**Figure 1 molecules-27-00938-f001:**
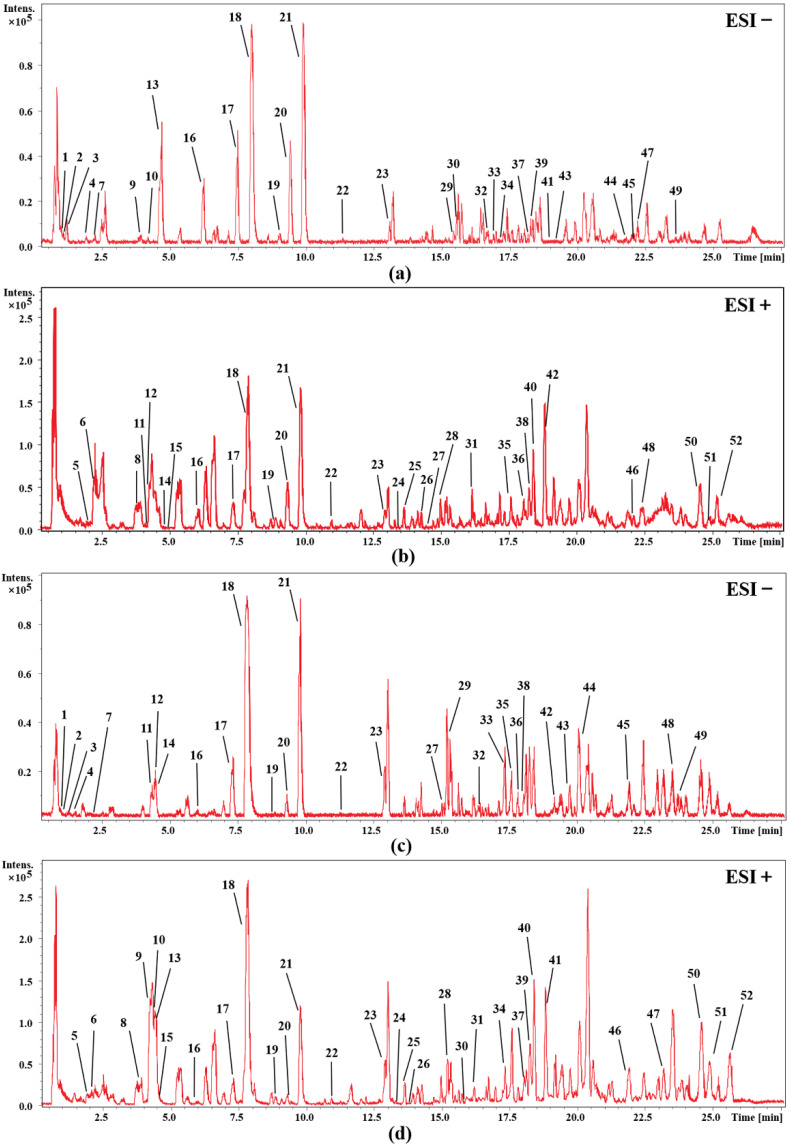
UHPLC-Q-TOF-MS base peak intensity chromatograms of *C. dahurica*. (**a**) rhizome of *C. dahurica* in negative mode; (**b**) rhizome of *C. dahurica* in positive mode; (**c**) fibrous roots of *C. dahurica* in negative mode; (**d**) fibrous roots of *C. dahurica* in positive mode.

**Figure 2 molecules-27-00938-f002:**
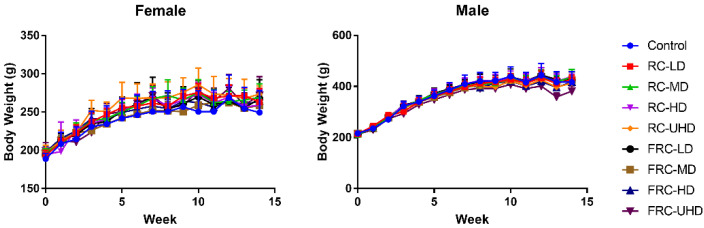
Effects of rhizome and fibrous roots of *C. dahurica* on the body weight changes after oral administration in male and female rats for 90 days: 0.585 g/kg (LD), 1.755 g/kg (MD), 5.265 g/kg (HD) and 15.795 g/kg (UHD). Data expressed as means ± SD.

**Figure 3 molecules-27-00938-f003:**
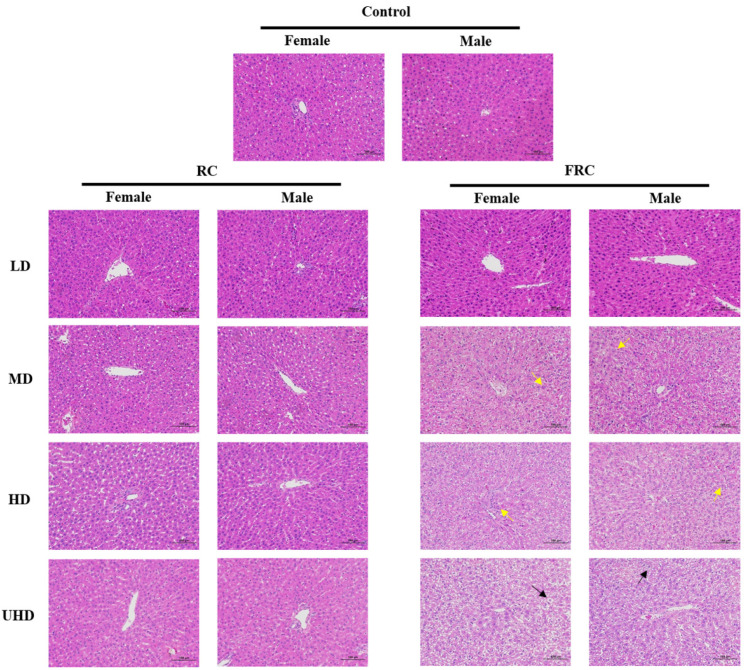
Effects of rhizome and fibrous roots of *C. dahurica* on histopathological changes after oral administration in male and female rats for 90 days. Hematoxylin–Eosin stain, 200×.

**Figure 4 molecules-27-00938-f004:**
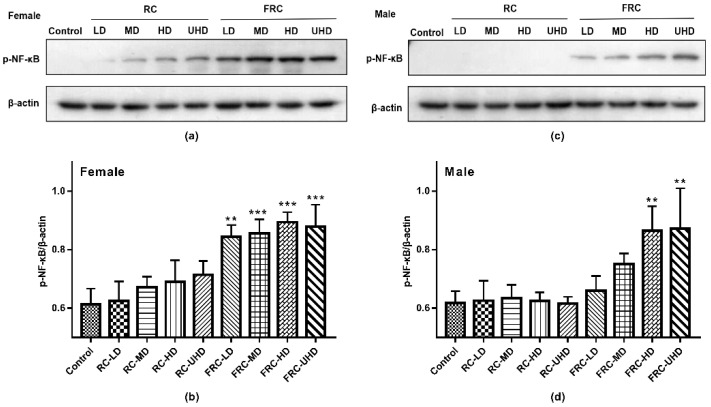
Effect of rhizome and fibrous roots of *C. dahurica* on the expression of p-NF-κB Protein in the liver after oral administration in male and female rats for 90 days. (**a**) p-NF-κB and β-actin protein electrophoresis images of female rats. (**b**) p-NF-κB/β-actin column charts of female rats. (**c**) p-NF-κB and β-actin protein electrophoresis images of male rats. (**d**) p-NF-κB/β-actin column charts of male rats. Data were expressed as means ± SD (n = 3). ** *p* < 0.01; *** *p* < 0.001.

**Table 1 molecules-27-00938-t001:** Characterization of compounds identified from rhizome of *C. dahurica* by UHPLC-Q-TOF-MS.

NO.	t_R_ (min)	Measured Mass (*m*/*z*)	Extraction Mass (*m*/*z*)	Formula	Error (ppm)	Fragment Ions (*m*/*z*)	Identified Compounds
1	1.10	271.0454 [M − H]^−^	271.0459	C_11_H_12_O_8_	1.9	253.0340, 181.0497, 123.0458	Fukiic acid
2	1.21	315.1079 [M − H]^−^	315.1085	C_14_H_20_O_8_	2.1	153.0566, 123.0451	Cimidahurinine
3	1.24	255.0506 [M − H]^−^	255.0510	C_11_H_12_O_7_	1.8	193.0487, 179.0334, 165.0553	Piscidic acid
4	1.85	315.1097 [M − H]^−^	315.1085	C_14_H_20_O_8_	−3.7	153.0542, 123.0441	Cimidahurine
5	2.08	379.0985 [M + Na]^+^	379.1000	C_16_H_20_O_9_	3.8	177.0530	cis-Ferulic acid-4-*O*-β-d-gal
6	2.12	379.0982 [M + Na]^+^	379.1000	C_16_H_20_O_9_	4.8	177.0529	trans-Ferulic acid-4-*O*-β-d-gal
7	2.19	179.0348 [M − H]^−^	179.0350	C_9_H_8_O_4_	0.9	135.0454	Caffeic acid
8	4.09	476.1902 [M + H]^+^	476.1915	C_24_H_29_NO_9_	2.7	314.1370, 177.0524	trans-feruloyl tyramine-4-*O*-β-d-glu
9	4.12	193.0501 [M − H]^−^	193.0506	C_10_H_10_O_4_	2.9	178.0231, 134.0359	Ferulic acid
10	4.17	193.0509 [M − H]^−^	193.0506	C_10_H_10_O_4_	−1.2	178.0244, 134.0368	Isoferulic acid
11	4.23	307.1163 [M + H]^+^	307.1176	C_16_H_18_O_6_	4.2	289.1053, 259.0590, 235.0583	Cimifugin
12	4.36	457.0719 [M + Na]^+^	457.0741	C_20_H_18_O_11_	4.8	295.0396, 163.0372	Cimicifugic acid G
13	4.73	433.0769 [M − H]^−^	433.0776	C_20_H_18_O_11_	1.7	271.0450, 179.0344, 135.0452	Fukinolic acid
14	4.81	506.2011 [M + H]^+^	506.2021	C_25_H_31_NO_10_	1.9	344.1466, 177.0532	Isocimicifugamide
15	4.83	506.2003 [M + H]^+^	506.2021	C_25_H_31_NO_10_	3.5	344.1456, 177.0530	Cimicifugamide
16	6.07	441.0755 [M + Na]^+^	411.0792	C_20_H_18_O_10_	3.4	279.0453, 163.0378	Cimicifugic acid D
	6.25	417.0813 [M − H]^−^	417.0827		3.5	179.0342, 135.0447,	
17	7.39	471.0865 [M + Na]^+^	471.0898	C_21_H_20_O_11_	3.3	295.0390, 177.0525	Cimicifugic acid A
	7.47	447.0916 [M − H]^−^	447.0933		3.9	253.0341, 181.0499, 165.0556, 109.0286	
18	7.94	471.0864 [M + Na]^+^	471.0898	C_21_H_20_O_12_	3.3	295.0398, 177.0528	Cimicifugic acid B
	8.01	447.0912 [M − H]^−^	447.0933		4.6	253.0343, 181.0496, 165.0544, 109.0288	
19	8.92	485.1025 [M + Na]^+^	485.1054	C_22_H_22_O_11_	3.0	279.0461, 207.0623	2-isoferuloyl fukinolic acid-1-melyl esier
	9.12	461.1083 [M − H]^−^	461.1089		1.4	253.0601, 193.0499, 165.0547, 149.0607	
20	9.39	455.0918 [M + Na]^+^	455.0949	C_21_H_20_O_10_	3.1	279.0453, 177.0533	2-feruloyl-piscidic acid
	9.55	431.0974 [M − H]^−^	431.0984		2.3	237.0396, 193.0494, 165.0544	
21	9.86	431.0969 [M − H]^−^	431.0984	C_21_H_20_O_10_	3.5	237.0395, 193.0491, 165.0543	2-isoferuloyl-piscidic acid
	9.88	455.0919 [M + Na]^+^	455.0949		3.0	279.0458, 177.0524	
22	10.99	485.1034 [M + Na]^+^	485.1054	C_22_H_22_O_11_	2.1	279.0432, 207.0601	2-feruloyl-fukinolic acid-1-melyl esier
	11.37	461.1085 [M − H]^−^	461.1089		0.9	253.0339, 181.0503, 109.0280	
23	12.94	367.076 [M + Na]^+^	367.0788	C_18_H_16_O_7_	2.8	299.4364, 177.0667	4′-methoxyl-3′-hydroxy-carboxy benzoyl isoferulic acid anhyelride
	13.07	343.0814 [M − H]^−^	343.0823		2.8	193.0497, 134.0364	
24	13.52	657.3595 [M + Na]^+^	657.3609	C_35_H_54_O_10_	2.2	617.3627, 467.3116	12β-*O*-7,8-Didehydroxycimigenol-3-*O*-β-d-xyl
25	13.59	659.3772 [M + Na]^+^	659.3766	C_35_H_56_O_10_	−0.9	619.3786, 487.3391, 469.3289	(22*R*)-22β-Hydroxycimigenol-3-*O*-β-d-xyl
26	14.32	703.3993 [M + Na]^+^	703.4028	C_37_H_60_O_11_	5.0	663.4060, 645.3944, 513.3543, 495.3408, 453.3217	24-epi-*O*-Acetylhydrosheng manol-3-*O*-β-ara
27	14.47	701.3843 [M + Na]^+^	701.3871	C_37_H_58_O_11_	4.0	679.4008, 661.3910, 601.3685, 451.3176	12β-Acetylcimigenol-3-*O*-β-d-xyl
28	14.99	745.3728 [M + Na]^+^	745.3770	C_38_H_58_O_13_	4.5	745.3728	Cimilactone K
29	15.57	198.0916 [M − H]^−^	198.0924	C_13_H_13_NO	4.4	198.0916	(*E*)-3-(3-methyl-2-butenylidene)-2-indolinone
30	15.59	198.0922 [M − H]^−^	198.0924	C_13_H_13_NO	1.3	198.0922	(*Z*)-3-(3-methyl-2-butenylidene)-2-indolinone
31	16.16	787.3830 [M + Na]^+^	787.3875	C_40_H_60_O_14_	4.8	585.3597, 435.3159	24-epi-24-*O*-Acetyl-7,8-didehydroshengmanol-3-*O*-(2′-*O*-malonyl)-β-d-xyl
32	16.70	635.3796 [M − H]^−^	635.3801	C_35_H_56_O_10_	0.8	577.3356	7β-Hydroxycimigenol-3-*O*-β-D-xyl
33	16.92	637.3948 [M − H]^−^	637.3957	C_35_H_58_O_10_	1.5	579.3510	Beesioside E
34	17.09	679.4091 [M − H]^−^	679.4063	C_37_H_60_O_11_	−4.1	619.3796	24-*O*-acetylhydro-shengmanol-3-*O*-β-d-xyl
35	17.61	643.3798 [M + Na]^+^	643.3817	C_35_H_56_O_9_	1.8	585.3713, 435.3190	Cimigenol-3-*O*-β-d-xyl (Cimigenoside)
36	18.06	683.3700 [M + Na]^+^	683.3766	C_37_H_56_O_10_	4.9	643.3609, 435.3288	25-*O*-Acetyl-7,8-didehydrocimigenol-3-*O*-β-d-xyl
37	18.08	635.3789 [M − H]^−^	635.3801	C_35_H_56_O_10_	1.8	577.3359	12β-hydroxycimigenol-3-*O*-α-L-ara
38	18.26	643.3790 [M + Na]^+^	643.3817	C_35_H_56_O_9_	2.7	585.3726, 435.3295	Cimigenol-3-*O*-α-L-ara (Cimiracemoside C)
39	18.30	637.3957 [M − H]^−^	637.3957	C_35_H_58_O_10_	0.1	579.3521	Beesioside B
40	18.40	703.3967 [M + Na]^+^	703.4028	C_37_H_60_O_11_	2.7	645.3944, 513.3544	24-epi-24-*O*-Acetylhydroshengmanol-3-*O*-β-d-xyl
41	18.76	679.4069 [M − H]^−^	679.4063	C_37_H_60_O_11_	−1.0	619.3770	24-*O*-acetylhydroshengmanol-3-*O*-β-D-xyl
42	18.83	701.3828 [M + Na]^+^	701.3871	C_37_H_58_O_11_	4.5	451.3172, 274.2716	24-*O*-Acetyl-7,8-didehydro-hydroshengmanol-3-*O*-β-d-xyl
43	19.11	707.3979 [M − H]^−^	707.4012	C_38_H_60_O_12_	4.7	661.3903, 619.3691, 469.3570	24-Epi-24-*O*-acetyl-7,8-dehydro cohosh alcohol-3-*O*-β-d-gal
44	21.77	659.3785 [M − H]^−^	659.3801	C_37_H_56_O_10_	2.4	617.3720, 559.3198	27-Deoxy Arcot hormone
45	21.97	661.3938 [M − H]^−^	661.3957	C_37_H_58_O_10_	2.9	619.3715, 601.3583	23-*O*-Acetyl alcohol cimicifuga-3-*O*-β-xyl
46	22.11	511.3374 [M + Na]^+^	511.3394	C_30_H_48_O_5_	1.9	453.3342	24-epi-Cimigenol
47	22.18	677.3889 [M − H]^−^	677.3906	C_37_H_58_O_11_	2.6	617.3732	7,8-Deoxy cohosh alcohol-24-*O*-acetylalcohol-ara
48	22.45	511.3368 [M + Na]^+^	511.3394	C_30_H_48_O_5_	2.6	435.3219	Cimigenol
49	23.56	665.3882 [M − H]^−^	665.3906	C_36_H_58_O_11_	3.7	619.3203	12β-Hydroxy cohosh alcohol-3-*O*-β-d-gal
50	24.56	863.4360 [M + Na]^+^	863.4400	C_43_H_68_O_16_	3.9	803.7323, 643.3770, 572.4235, 435.3229	Heracleifolinoside F
51	24.82	569.3434 [M + Na]^+^	569.3449	C_32_H_50_O_7_	1.5	529.3466, 511.3389	24-*O*-Acetyl-7,8-didehydro-hydroshengmanol
52	25.12	571.3590 [M + Na]^+^	571.3605	C_32_H_52_O_7_	1.5	513.3530, 453.3352	24-*O*-Acetylhydroshengmanol

**Table 2 molecules-27-00938-t002:** Characterization of compounds identified from fibrous roots of *C. dahurica* by UHPLC-Q-TOF-MS.

NO.	t_R_ (min)	Measured Mass (*m*/*z*)	Extraction Mass (*m*/*z*)	Formula	Error (ppm)	Fragment Ions (*m*/*z*)	Identified Compounds
1	1.07	271.0457 [M − H]^−^	271.0459	C_11_H_12_O_8_	0.8	253.0379, 181.0493, 123.0428	Fukiic acid
2	1.12	315.1079 [M − H]^−^	315.1085	C_14_H_20_O_8_	2.0	153.0553, 123.0437	Cimidahurinine
3	1.16	255.0507 [M − H]^−^	255.0510	C_11_H_12_O_7_	1.8	193.0467, 179.0293, 165.0559	Piscidic acid
4	1.19	315.1082 [M − H]^−^	315.1085	C_14_H_20_O_8_	1.1	153.0560, 123.0433	Cimidahurine
5	2.00	379.0991 [M + Na]^+^	379.1000	C_16_H_20_O_9_	2.4	177.0541	cis-Ferulic acid-4-*O*-β-d-gal
6	2.15	379.0990 [M + Na]^+^	379.1000	C_16_H_20_O_9_	2.6	177.0537	trans-Ferulic acid-4-*O*-β-d-gal
7	2.17	179.0350 [M − H]^−^	179.0350	C_9_H_8_O_4_	−0.1	135.0431	Caffeic acid
8	4.00	476.1899 [M + H]^+^	476.1915	C_24_H_29_NO_9_	3.5	314.1380, 177.0548	trans-feruloyl tyramine-4-*O*-β-d-glu
9	4.33	457.0734 [M + Na]^+^	457.0741	C_20_H_18_O_11_	1.7	295.0406, 163.0382	Cimicifugic acid G
10	4.39	307.1167 [M + H]^+^	307.1176	C_16_H_18_O_6_	3.0	289.1044, 259.0595, 235.0592	Cimifugin
11	4.39	433.0758 [M − H]^−^	433.0776	C_20_H_18_O_11_	4.3	271.0476, 179.0342, 135.0449	Fukinolic acid
12	4.40	193.0503 [M − H]^−^	193.0506	C_10_H_10_O_4_	1.7	178.0308, 134.0359	Ferulic acid
13	4.46	506.2009 [M + H]^+^	506.2021	C_25_H_31_NO_10_	2.2	344.1520, 177.0540	Isocimicifugamide
14	4.52	193.0497 [M − H]^−^	193.0506	C_10_H_10_O_4_	4.7	178.0261, 134.0380	Isoferulic acid
15	4.69	506.2006 [M + H]^+^	506.2021	C_25_H_31_NO_10_	2.8	344.1492, 177.0557	Cimicifugamide
16	6.06	441.0783 [M + Na]^+^	411.0792	C_20_H_18_O_10_	2.2	279.0482,163.0377	Cimicifugic acid D
	6.07	417.0821 [M − H]^−^	417.0827		1.5	255.0465, 193.0519, 179.0347	
17	7.27	447.0920 [M − H]^−^	447.0933	C_21_H_20_O_11_	2.8	253.0341, 181.0495, 165.0538, 109.0294	Cimicifugic acid A
	7.37	471.0896 [M + Na]^+^	471.0898		0.4	295.0435, 177.0549	
18	7.87	471.0895 [M + Na]^+^	471.0898	C_21_H_20_O_12_	0.5	295.0417, 177.0548	Cimicifugic acid B
	7.88	447.0918 [M − H]^−^	447.0933		3.3	253.0348, 181.0500, 165.0550, 109,0291	
19	8.91	461.1077 [M − H]^−^	461.1089	C_22_H_22_O_11_	2.7	165.0536, 193.0495, 233.0624	2-isoferuloyl fukinolic acid-1-melyl esier
	8.92	485.1059 [M + Na]^+^	485.1054		−1.0	279.0494, 207.0615	
20	9.35	455.0941 [M + Na]^+^	455.0949	C_21_H_20_O_10_	1.6	367.0784, 279.0461	2-feruloyl-piscidic acid
	9.40	431.0978 [M − H]^−^	431.0984		1.4	237.0418, 193.0560, 165.0560	
21	9.74	431.0964 [M − H]^−^	431.0984	C_21_H_20_O_10_	4.6	237.0398, 193.0500, 165.0548	2-isoferuloyl-piscidic acid
	9.87	455.0947 [M + Na]^+^	455.0949		0.3	279.0473, 177.0547	
22	11.29	485.1047 [M + Na]^+^	485.1054	C_22_H_22_O_11_	1.5	279.0456, 207.0672	2-feruloyl-fukinolic acid-1-melyl esier
	11.41	461.1072 [M − H]^−^	461.1089		3.8	253.0352, 181.0496, 109.0289	
23	12.90	343.0809 [M − H]^−^	343.0823	C_18_H_16_O_7_	4.1	193.0499, 134.0363	4′-methoxyl-3′-hydroxy-carboxy benzoyl isoferulic acid anhyelride
	12.92	367.0768 [M + Na]+	367.0788		2.7	299.4364,177.0667	
24	13.48	657.3593 [M + Na]^+^	657.3609	C_35_H_54_O_10_	2.4	617.3645, 467.3129	12β-*O*-7,8-Didehydroxycimigenol-3-*O*-β-d-xyl
25	13.70	659.3736 [M + Na]^+^	659.3766	C_35_H_56_O_10_	4.5	451.3189, 177.0521	(22*R*)-22β-Hydroxycimigenol-3-*O*-β-d-xyl
26	13.76	703.4055 [M + Na]^+^	703.4028	C_37_H_60_O_11_	−3.9	663.4041, 645.3946, 513.3544, 495.3244, 453.3280	24-epi-*O*-Acetylhydrosheng manol-3-*O*-β-ara
27	15.18	198.0920 [M − H]^−^	198.0924	C_13_H_13_NO	2.5	198.0913	(*E*)-3-(3-methyl-2-butenylidene)-2-indolinone
28	15.24	701.3838 [M + Na]^+^	701.3871	C_37_H_58_O_11_	4.8	679.4007, 661.3891, 601.3702, 451.3158	12β-Acetylcimigenol-3-*O*-β-d-xyl
29	15.26	198.0924 [M − H]^−^	198.0924	C_13_H_13_NO	0.4	198.0924	(*Z*)-3-(3-methyl-2-butenylidene)-2-indolinone
30	15.89	745.3764 [M + Na]^+^	745.3770	C_38_H_58_O_13_	0.7	671.3720, 513.3649, 435.3291	Cimilactone K
31	16.24	787.3838 [M + Na]^+^	787.3875	C_40_H_60_O_14_	4.7	647.3384, 526.3150, 429.2978	24-epi-24-*O*-Acetyl-7,8-didehydroshengmanol-3-*O*-(2′-*O*-malonyl)-β-d-xyl
32	16.44	635.3794 [M − H]^−^	635.3801	C_35_H_56_O_10_	1.1	577.3356	7β-Hydroxycimigenol-3-*O*-β-D-xyl
33	17.30	637.3927 [M − H]^−^	637.3957	C_35_H_58_O_10_	4.7	579.3483	Beesioside E
34	17.37	643.3786 [M + Na]^+^	643.3817	C_35_H_56_O_9_	3.8	585.3712, 435.3242	Cimigenol-3-*O*-β-d-xyl (Cimigenoside)
35	17.52	679.4043 [M − H]^−^	679.4063	C_37_H_60_O_11_	2.9	619.377	24-*O*-acetylhydro-shengmanol-3-*O*-β-d-xyl
36	17.79	635.3777 [M − H]^−^	635.3801	C_35_H_56_O_10_	3.7	577.3352	12β-hydroxycimigenol-3-*O*-α-l-ara
37	18.06	683.3731 [M + Na]^+^	683.3766	C_37_H_56_O_10_	5.0	511.3385, 453.3369	25-*O*-Acetyl-7,8-didehydrocimigenol-3-*O*-β-d-xyl
38	18.09	637.3928 [M − H]^−^	637.3957	C_35_H_58_O_10_	4.6	579.3481	Beesioside B
39	18.28	643.3803 [M + Na]^+^	643.3817	C_35_H_56_O_9_	2.0	585.3746, 435,3225	Cimigenol-3-*O*-α-l-ara (Cimiracemoside C)
40	18.42	703.4001 [M + Na]^+^	703.4028	C_37_H_60_O_11_	3.8	645.3962, 513.3558	24-epi-24-*O*-Acetylhydroshengmanol-3-*O*-β-d-xyl
41	18.81	701.3828 [M + Na]^+^	701.3871	C_37_H_58_O_11_	5.4	451.3210	24-*O*-Acetyl-7,8-didehydro-hydroshengmanol-3-*O*-β-d-xyl
42	19.17	679.4031 [M − H]^−^	679.4063	C_37_H_60_O_11_	4.7	619.3784	24-*O*-acetylhydroshengmanol-3-*O*-β-d-xyl
43	19.63	707.3988 [M − H]^−^	707.4012	C_38_H_60_O_12_	3.4	661.3891, 619.3727, 469.3341	24-Epi-24-*O*-acetyl-7,8-dehydro cohosh alcohol-3-*O*-β-D-gal
44	20.15	659.3770 [M − H]^−^	659.3801	C_37_H_56_O_10_	2.4	617.3614, 559.3276	27-Deoxy Arcot hormone
45	21.92	661.3937 [M − H]^−^	661.3957	C_37_H_58_O_11_	3.0	619.3892, 601.3476	23-*O*-Acetyl alcohol cimicifuga-3-*O*-β-xyl
46	21.96	511.3379 [M + Na]^+^	511.3394	C_30_H_48_O_5_	2.9	453.3355, 471.3932	24-epi-Cimigenol
47	23.19	511.3396 [M + Na]^+^	511.3394	C_30_H_48_O_5_	−0.5	453.3340, 435.3219	Cimigenol
48	23.39	677.3897 [M − H]^−^	677.3906	C_37_H_58_O_11_	1.4	617.3685	7,8-Deoxy cohosh alcohol-24-*O*-acetylalcohol-ara
49	23.61	665.3908 [M − H]^−^	665.3906	C_36_H_58_O_11_	−0.3	619.3825	12β-Hydroxy cohosh alcohol-3-*O*-β-d-gal
50	24.62	643.3783 [M + Na]^+^	643.3817	C_35_H_56_O_9_	5.2	529.3487, 453.3347	Cimiaceroside B
51	24.92	569.3432 [M + Na]^+^	569.3449	C_32_H_50_O_7_	2.9	529.3481, 511.3442	24-*O*-Acetyl-7,8-didehydro-hydroshengmanol
52	25.55	571.3580 [M + Na]^+^	571.3605	C_32_H_52_O_7_	4.4	531.3659, 453.3344	24-*O*-Acetylhydroshengmanol

**Table 3 molecules-27-00938-t003:** Urinalysis of rats with 90-day repeated administration of rhizome and fibrous roots of *C. dahurica*.

Group	Control	RC				FRC			
	0	LD	MD	HD	UHD	LD	MD	HD	UHD
Females									
WBC (Cell/μL)	-	-	-	-	-	-	-	-	-
KET (mmol/L)	-	-	-	-	-	-	-	-	-
NIT	-	-	-	-	-	-	-	-	-
URO (μmol/L)	Normal	Normal	Normal	Normal	Normal	Normal	Normal	Normal	Normal
BIL (μmol/L)	-	-	-	-	-	-	-	-	-
PRO (g/L)	+3	+3	+2	+3	+3	+3	+3	+3	+2
GLU	-	-	-	-	-	-	-	-	-
SG	1.005	1.005	1.010	1.005	1.005	1.005	1.010	1.010	1.020
BLD (Cell/μL)	-	-	-	-	-	-	-	-	-
pH	7.5	7.5	7.5	7.5	7.5	7.5	7.5	7.5	7.0
Vc (mmol/L)	±	±	-	±	±	±	±	±	±
**Males**									
WBC (Cell/μL)	-	-	-	-	-	-	-	-	-
KET (mmol/L)	-	-	-	-	-	-	-	-	-
NIT	-	-	-	-	-	-	-	-	-
URO (μmol/L)	Normal	Normal	Normal	Normal	Normal	Normal	Normal	Normal	Normal
BIL (μmol/L)	-	-	-	-	-	-	-	-	-
PRO (g/L)	+3	+3	+3	+3	+3	+2	+3	+3	+2
GLU	-	-	-	-	-	-	-	-	-
SG	1.005	1.005	1.010	1.005	1.005	1.005	1.005	1.010	1.015
BLD (Cell/μL)	-	-	-	-	-	-	-	-	-
pH	7.5	7.0	7.5	7.5	7.5	7.0	7.5	7.0	7.0
Vc (mmol/L)	±	±	±	±	±	±	±	±	±

Rhizome extract (RC), fibrous roots extract (FRC), 0.585 g/kg (LD), 1.755 g/kg (MD), 5.265 g/kg (HD), 15.795 g/kg (UHD), White blood cells (WBC), ketone bodies (KET), nitrite (NIT), urine bilirubin (URO), bil-irubin (BIL), protein (PRO), glucose (GLU), specific gravity (SG), hemoglobin (BLD) and Vitamin C (Vc).

**Table 4 molecules-27-00938-t004:** Hematological analysis of rats with 90-day repeated administration of rhizome and fibrous roots of *C. dahurica*.

Group	Control	RC				FRC			
	0	LD	MD	HD	UHD	LD	MD	HD	UHD
Females									
WBC (10^9^ cells/L)	7.4 ± 0.3	6.8 ± 0.6	7.2 ± 1.0	7.9 ± 1.4	5.2 ± 0.5	10.5 ± 1.3 *	10.3 ± 0.6 *	10.4 ± 0.7 **	10.9 ± 1.2 **
RBC (10^12^ cells/L)	6.9 ± 0.7	6.2 ± 0.3	6.4 ± 0.5	6.4 ± 0.6	6.5 ± 0.2	6.7 ± 1.0	6.1 ± 0.3	6.9 ± 0.4	6.5 ± 1.1
HGB (g/L)	131.7 ± 8.3	130.5 ± 8.6	129.2 ± 9.6	126.4 ± 10.4	124.8 ± 4.5	132.5 ± 9.7	124.3 ± 6.0	138.0 ± 3.7	128.0 ± 14.9
HCT (%)	43.6 ± 2.0	42.6 ± 1.9	43.1 ± 2.4	42.1 ± 2.7	42.0 ± 2.1	44.2 ± 2.4	42.1 ± 2.2	43.9 ± 2.4	44.6 ± 1.8
PLT (10^11^ cells/L)	13.2 ± 1.5	12.4 ± 0.8	14.3 ± 1.1	12.0 ± 1.2	13.9 ± 2.5	11.6 ± 2.4	11.6 ± 1.1	11.9 ± 1.8	11.9 ± 2.7
MCV (fL)	65.6 ± 2.0	68.4 ± 1.0	67.7 ± 1.5	66.3 ± 2.5	64.7 ± 2.6	66.3 ± 2.4	68.6 ± 1.8	67.0 ± 2.0	67.1 ± 2.2
MCH (pg)	19.2 ± 1.0	21.0 ± 0.3	20.3 ± 0.5	19.9 ± 0.4	19.3 ± 0.4	20.6 ± 0.7	20.3 ± 0.4	20.1 ± 0.7	19.7 ± 0.9
MCHC (g/L)	301.7 ± 5.5	306.0 ± 7.3	300.0 ± 6.9	300.0 ± 6.2	297.5 ± 5.3	299.8 ± 6.6	295.8 ± 3.9	299.2 ± 4.2	298.6 ± 5.0
Neutrophils (%)	11.1 ± 1.7	10.8 ± 4.4	11.3 ± 4.3	11.4 ± 2.6	14.2 ± 3.1	15.5 ± 5.2	15.9 ± 4.2	17.5 ± 5.5	18.9 ± 3.4
Eosinophils (%)	2.3 ± 0.2	2.6 ± 0.8	1.5 ± 0.2	2.2 ± 1.1	2.8 ± 0.4	3.4 ± 0.9	3.3 ± 1.0	2.2 ± 0.2	2.1 ± 0.3
Basophils (%)	0.5 ± 0.2	0.5 ± 0.1	0.4 ± 0.2	0.5 ± 0.3	0.5 ± 0.1	0.5 ± 0.1	0.3 ± 0.1	0.5 ± 0.2	0.4 ± 0.2
Lymphocytes (%)	76.8 ± 12.8	88.1 ± 3.8	86.0 ± 4.4	85.1 ± 3.2	71.2 ± 13.0	75.4 ± 10.2	78.9 ± 3.9	85.2 ± 6.6	83.3 ± 2.9
Monocytes (%)	0.9 ± 0.5	0.5 ± 0.1	0.7 ± 0.3	0.7 ± 0.2	0.7 ± 0.1	0.9 ± 0.1	0.8 ± 0.2	0.8 ± 0.2	0.7 ± 0.2
Reticulocytes (%)	12.1 ± 2.3	13.8 ± 1.7	12.0 ± 1.4	11.7 ± 2.5	10.7 ± 0.8	14.9 ± 1.7	10.4 ± 0.8	12.1 ± 1.8	11.5 ± 1.5
**Males**									
WBC (10^9^ cells/L)	7.4 ± 1.0	8.5 ± 1.2	8.8 ± 0.6	9.0 ± 0.9	8.9 ± 0.6	8.8 ± 1.0	9.0 ± 0.3	9.5 ± 0.5	10.4 ± 1.0 **
RBC (10^12^ cells/L)	6.8 ± 0.7	7.8 ± 0.5	7.9 ± 0.3	7.6 ± 0.4	7.5 ± 0.6	7.0 ± 1.4	7.4 ± 0.6	7.3 ± 0.7	7.7 ± 0.2
HGB (g/L)	146.0 ± 4.4	149.0 ± 5.3	143.7 ± 2.3	141.5 ± 3.7	137.3 ± 5.1	142.0 ± 4.6	141.7 ± 4.6	144.7 ± 5.8	141.7 ± 5.7
HCT (%)	42.1 ± 2.0	47.0 ± 2.6	47.8 ± 0.3	46.6 ± 1.8	45.9 ± 2.1	48.4 ± 3.1	47.6 ± 4.1	47.4 ± 3.9	47.4 ± 2.2
PLT (10^11^ cells/L)	12.0 ± 1.8	12.1 ± 1.5	13.0 ± 1.5	14.2 ± 1.7	11.7 ± 1.7	11.0 ± 1.5	12.1 ± 1.9	11.2 ± 1.1	11.2 ± 1.5
MCV (fL)	60.6 ± 1.3	60.5 ± 1.6	60.4 ± 2.4	61.0 ± 1.6	61.2 ± 2.0	64.2 ± 4.3	64.7 ± 4.3	64.8 ± 1.6	65.1 ± 3.9
MCH (pg)	18.6 ± 1.3	18.7 ± 0.4	18.2 ± 0.4	18.6 ± 0.6	18.4 ± 0.9	20.4 ± 1.7	19.7 ± 1.1	19.3 ± 0.5	18.6 ± 0.4
MCHC (g/L)	299.0 ± 5.6	309.2 ± 4.7	301.3 ± 6.4	309.3 ± 6.4	294.5 ± 5.2	308.5 ± 2.6	304.7 ± 5.0	297.7 ± 1.2	296.3 ± 3.5
Neutrophils (%)	14.6 ± 2.1	17.9 ± 3.3	16.2 ± 2.7	17.2 ± 1.7	17.8 ± 5.7	17.6 ± 3.8	15.1 ± 5.0	20.0 ± 7.2	19.3 ± 6.3
Eosinophils (%)	4.8 ± 1.1	3.4 ± 1.6	2.7 ± 1.3	3.1 ± 1.9	2.4 ± 1.2	2.7 ± 1.8	2.4 ± 0.6	3.7 ± 1.2	3.4 ± 1.1
Basophils (%)	0.3 ± 0.1	0.6 ± 0.2	0.4 ± 0.1	0.4 ± 0.1	0.5 ± 0.1	0.6 ± 0.1	0.6 ± 0.2	0.5 ± 0.4	0.4 ± 0.2
Lymphocytes (%)	74.5 ± 6.9	72.0 ± 12.3	79.7 ± 2.2	75.4 ± 6.3	76.1 ± 7.2	76.6 ± 5.8	80.7 ± 4.8	79.6 ± 5.5	80.6 ± 2.2
Monocytes (%)	1.0 ± 0.3	0.8 ± 0.5	0.9 ± 0.1	0.8 ± 0.3	1.1 ± 0.2	1.2 ± 1.0	1.0 ± 0.6	0.8 ± 0.2	0.8 ± 0.2
Reticulocytes (%)	6.8 ± 1.5	7.6 ± 0.8	7.8 ± 0.5	9.0 ± 2.2	9.6 ± 1.4	9.8 ± 1.9	9.7 ± 2.1	9.7 ± 0.7	9.9 ± 2.3

Total white blood cells (WBC), red blood cells (RBC), hemoglobin (HGB), hematocrit (HCT), platelets (PLT), mean corpuscular volume (MCV), mean corpuscular hemoglobin (MCH) and mean corpuscular hemoglobin concentration (MCHC). * Significantly different from Control group (*p* < 0.05). ** Significantly different from Control group (*p* < 0.01).

**Table 5 molecules-27-00938-t005:** Biochemical analysis of rats with 90-day repeated administration of rhizome and fibrous roots of *C. dahurica*.

Group	Control	RC				FRC			
	0	LD	MD	HD	UHD	LD	MD	HD	UHD
Females									
ALP (U/L)	38.2 ± 9.0	33.7 ± 10.6	35.9 ± 11.9	36.3 ± 11.8	34.3 ± 12.4	39.8 ± 10.3	32.4 ± 6.5	39.9 ± 7.2	41.0 ± 11.1
ALT (U/L)	34.8 ± 2.5	42.5 ± 4.8	42.7 ± 3.7	42.4 ± 5.4	42.8 ± 5.2	46.0 ± 6.2 **	39.3 ± 2.8	43.0 ± 6.0 *	63.1 ± 8.4 ***
AST (U/L)	182.1 ± 28.1	168.9 ± 17.7	174.5 ± 23.2	208.5 ± 56.7	215.7 ± 42.4	248.0 ± 42.4 *	237.7 ± 41.4 *	227.3 ± 67.5	185.5 ± 20.5
UREA (mmol/L)	6.7 ± 0.5	6.2 ± 0.4	6.5 ± 0.6	7.1 ± 0.4	6.9 ± 0.4	7.3 ± 0.7	8.0 ± 0.7 **	7.4 ± 0.7	7.5 ± 0.9
CREA (Umol/L)	25.8 ± 2.2	26.9 ± 6.2	25.2 ± 2.4	26.5 ± 2.0	25.9 ± 4.5	28.1 ± 4.3	27.4 ± 6.8	27.6 ± 2.0	29.0 ± 6.5
TBIL (Umol/L)	5.8 ± 1.7	5.4 ± 1.4	6.1 ± 1.4	5.7 ± 1.0	6.4 ± 1.1	6.1 ± 1.4	5.7 ± 0.9	5.9 ± 1.4	5.6 ± 1.4
GLU (mmol/L)	3.0 ± 0.2	3.3 ± 0.7	3.3 ± 0.4	3.3 ± 0.4	2.8 ± 0.3	3.0 ± 0.6	2.9 ± 0.5	2.8 ± 0.7	3.6 ± 0.5
CHOL (mmol/L)	2.0 ± 0.2	1.8 ± 0.2	1.8 ± 0.2	1.9 ± 0.1	1.9 ± 0.3	2.0 ± 0.2	2.2 ± 0.2	2.2 ± 0.2	2.5 ± 0.3 **
TG (mmol/L)	0.6 ± 0.1	0.5 ± 0.1	0.5 ± 0.1	0.6 ± 0.1	0.6 ± 0.0	0.6 ± 0.0	0.6 ± 0.1	0.6 ± 0.1	0.5 ± 0.0
TP (g/L)	60.7 ± 4.7	57.7 ± 2.8	61.0 ± 5.3	63.0 ± 2.6	63.2 ± 3.7	63.7 ± 2.5	63.1 ± 1.9	65.4 ± 3.0	69.4 ± 5.0
ALB (g/L)	39.7 ± 2.7	40.4 ± 3.5	40.8 ± 2.5	41.0 ± 1.2	41.4 ± 2.3	41.7 ± 0.9	40.7 ± 0.9	41.8 ± 1.4	44.7 ± 2.7
**Males**									
ALP (U/L)	58.9 ± 9.7	69.8 ± 12.4	66.8 ± 14.3	71.0 ± 8.2	57.1 ± 11.3	68.2 ± 17.7	58.4 ± 8.8	64.3 ± 9.6	90.8 ± 3.4 ***
ALT (U/L)	42.0 ± 7.0	55.1 ± 8.4	48.1 ± 6.1	52.5 ± 2.0	45.5 ± 6.0	54.9 ± 9.3	48.6 ± 4.7	48.5 ± 8.0	80.4 ± 11.9 ***
AST (U/L)	201.2 ± 11.4	241.5 ± 71.7	202.1 ± 33.7	234.8 ± 51.0	148.1 ± 54.8	262.7 ± 35.8	216.2 ± 32.4	192.1 ± 48.4	194.3 ± 35.7
UREA (mmol/L)	5.4 ± 0.3	6.2 ± 0.7	6.0 ± 0.4	6.2 ± 0.4	6.0 ± 0.7	7.2 ± 0.4 ***	6.3 ± 0.7 *	6.3 ± 0.7	7.5 ± 0.9 ***
CREA (Umol/L)	16.3 ± 3.6	15.1 ± 3.3	17.4 ± 3.3	20.8 ± 4.2	21.7 ± 4.4	21.7 ± 3.5	25.0 ± 3.5 **	19.5 ± 3.7	20.1 ± 2.6
TBIL (Umol/L)	5.2 ± 0.6	5.6 ± 0.8	5.4 ± 1.1	5.8 ± 1.3	5.7 ± 1.1	6.1 ± 0.6	6.6 ± 1.0	6.4 ± 1.1	7.6 ± 1.7 *
GLU (mmol/L)	4.0 ± 0.7	2.9 ± 0.8	3.4 ± 0.4	3.5 ± 0.7	3.3 ± 0.7	2.9 ± 0.7	3.6 ± 0.2	3.6 ± 0.6	3.5 ± 0.5
CHOL (mmol/L)	1.6 ± 0.2	1.7 ± 0.3	1.8 ± 0.2	1.8 ± 0.1	1.7 ± 0.2	1.8 ± 0.1	1.9 ± 0.2	1.8 ± 0.3	1.9 ± 0.3 *
TG (mmol/L)	0.5 ± 0.1	0.5 ± 0.1	0.4 ± 0.1	0.6 ± 0.1	0.5 ± 0.1	0.7 ± 0.1	0.6 ± 0.1	0.5 ± 0.1	0.4 ± 0.0
TP (g/L)	57.3 ± 5.8	64.1 ± 3.7	65.5 ± 5.6	63.7 ± 1.4	60.6 ± 3.1	63.5 ± 5.2	66.0 ± 4.1	67.3 ± 5.3	72.4 ± 4.8
ALB (g/L)	36.4 ± 1.8	40.9 ± 1.6	41.5 ± 2.5	40.0 ± 1.1	39.0 ± 1.5	40.3 ± 2.4	41.4 ± 1.7	42.9 ± 2.9	45.2 ± 1.5

Alkaline phosphatase (ALP), alanine aminotransferase (ALT), aspartate aminotransferase (AST), blood urea nitrogen (BUN), creatinine (CREA), total bilirubin (TBIL), glucose (GLU), total cholesterol (TC), triglycerides (TG), total protein (TP) and albumin (ALB). * Significantly different from Control group (*p* < 0.05). ** Significantly different from Control group (*p* < 0.01). *** Significantly different from Control group (*p* < 0.001).

**Table 6 molecules-27-00938-t006:** Organ weights and Organ index of rats with 90-day repeated administration of rhizome and fibrous roots of *C. dahurica*.

Group		Control	RH				FRC			
		0	LD	MD	HD	UHD	LD	MD	HD	UHD
Females										
Heart	(g)	0.9 ± 0.12	0.88 ± 0.06	0.88 ± 0.09	0.89 ± 0.07	1.19 ± 0.35	1.04 ± 0.15	0.91 ± 0.08	1.00 ± 0.20	1.05 ± 0.20
	(%BW)	0.36 ± 0.02	0.33 ± 0.02	0.33 ± 0.05	0.34 ± 0.01	0.38 ± 0.07	0.39 ± 0.08	0.35 ± 0.04	0.38 ± 0.08	0.38 ± 0.08
Liver	(g)	5.56 ± 0.58	5.56 ± 0.49	5.25 ± 0.27	7.55 ± 0.44	8.87 ± 1.28	7.90 ± 1.57	7.99 ± 1.10	7.89 ± 1.15	7.66 ± 1.10
	(%BW)	2.24 ± 0.17	2.12 ± 0.17	1.96 ± 0.18	2.93 ± 0.24	3.21 ± 0.46	2.87 ± 0.40	3.09 ± 0.38	2.99 ± 0.41	2.81 ± 0.46
Spleen	(g)	0.43 ± 0.06	0.50 ± 0.10	0.46 ± 0.07	0.59 ± 0.07	0.53 ± 0.12	0.45 ± 0.10	0.53 ± 0.08	0.49 ± 0.09	0.55 ± 0.07
	(%BW)	0.17 ± 0.01	0.19 ± 0.05	0.17 ± 0.02	0.23 ± 0.04	0.19 ± 0.05	0.17 ± 0.03	0.21 ± 0.04	0.19 ± 0.03	0.20 ± 0.03
Lung	(g)	1.27 ± 0.47	1.16 ± 0.13	1.02 ± 0.11	1.03 ± 0.11	1.20 ± 0.07	1.48 ± 0.36	1.14 ± 0.05	1.41 ± 0.60	1.23 ± 0.15
	(%BW)	0.52 ± 0.22	0.44 ± 0.05	0.38 ± 0.05	0.40 ± 0.05	0.44 ± 0.03	0.56 ± 0.15	0.44 ± 0.04	0.53 ± 0.22	0.45 ± 0.07
Kidney	(g)	1.72 ± 0.16	1.64 ± 0.19	1.57 ± 0.14	1.74 ± 0.19	2.04 ± 0.20	1.96 ± 0.20	1.28 ± 0.72	1.87 ± 0.14	1.86 ± 0.24
	(%BW)	0.70 ± 0.07	0.62 ± 0.06	0.58 ± 0.05	0.67 ± 0.04	0.74 ± 0.07	0.74 ± 0.10	0.66 ± 0.04	0.71 ± 0.03	0.68 ± 0.09
Stomach	(g)	1.59 ± 0.27	1.35 ± 0.10	1.49 ± 0.11	1.37 ± 0.25	1.48 ± 0.27	1.54 ± 0.15	1.43 ± 0.12	1.48 ± 0.09	1.63 ± 0.26
	(%BW)	0.64 ± 0.07	0.51 ± 0.04	0.56 ± 0.06	0.53 ± 0.09	0.54 ± 0.09	0.58 ± 0.05	0.55 ± 0.03	0.56 ± 0.04	0.59 ± 0.07
Brain	(g)	1.67 ± 0.15	1.77 ± 0.26	1.74 ± 0.09	1.79 ± 0.16	1.74 ± 0.17	1.48 ± 0.56	1.62 ± 0.14	1.74 ± 0.16	1.72 ± 0.21
	(%BW)	0.68 ± 0.11	0.67 ± 0.10	0.65 ± 0.07	0.69 ± 0.08	0.63 ± 0.08	0.62 ± 0.10	0.63 ± 0.04	0.66 ± 0.04	0.63 ± 0.08
Adrenal gland	(g)	0.18 ± 0.27	0.08 ± 0.01	0.07 ± 0.01	0.07 ± 0.01	0.10 ± 0.03	0.08 ± 0.01	0.08 ± 0.01	0.08 ± 0.01	0.09 ± 0.03
	(%BW)	0.03 ± 0.00	0.03 ± 0.00	0.03 ± 0.01	0.03 ± 0.00	0.04 ± 0.01	0.03 ± 0.01	0.03 ± 0.00	0.03 ± 0.00	0.03 ± 0.01
Thymus	(g)	0.23 ± 0.12	0.22 ± 0.06	0.21 ± 0.04	0.17 ± 0.04	0.21 ± 0.07	0.24 ± 0.12	0.17 ± 0.05	0.16 ± 0.03	0.25 ± 0.05
	(%BW)	0.09 ± 0.04	0.08 ± 0.02	0.08 ± 0.02	0.07 ± 0.01	0.08 ± 0.03	0.09 ± 0.05	0.06 ± 0.02	0.06 ± 0.01	0.09 ± 0.02
Genitals	(g)	0.63 ± 0.17	0.69 ± 0.20	0.46 ± 0.08	0.56 ± 0.09	0.70 ± 0.28	0.82 ± 0.39	0.46 ± 0.05	0.59 ± 0.22	0.67 ± 0.12
	(%BW)	0.25 ± 0.06	0.26 ± 0.07	0.17 ± 0.03	0.22 ± 0.02	0.25 ± 0.10	0.31 ± 0.16	0.18 ± 0.03	0.22 ± 0.08	0.24 ± 0.05
**Males**										
Heart	(g)	1.31 ± 0.10	1.29 ± 0.10	1.38 ± 0.15	9.51 ± 0.81	1.32 ± 0.12	1.35 ± 0.10	1.34 ± 0.14	1.36 ± 0.08	1.33 ± 0.24
	(%BW)	0.32 ± 0.03	0.30 ± 0.01	0.32 ± 0.03	0.28 ± 0.03	0.32 ± 0.04	0.31 ± 0.01	0.32 ± 0.02	0.26 ± 0.14	0.32 ± 0.05
Liver	(g)	7.93 ± 0.53	7.77 ± 0.65	7.49 ± 1.02	9.51 ± 0.81	10.68 ± 1.44	10.19 ± 2.23	11.13 ± 1.01	10.53 ± 0.81	12.03 ± 2.93
	(%BW)	1.91 ± 0.07	1.82 ± 0.15	1.72 ± 0.16	2.17 ± 0.13	2.59 ± 0.43	2.33 ± 0.35	2.65 ± 0.16	2.52 ± 0.11	2.92 ± 0.55
Spleen	(g)	0.54 ± 0.04	0.56 ± 0.10	0.63 ± 0.10	0.59 ± 0.09	0.67 ± 0.12	0.68 ± 0.19	0.64 ± 0.12	0.64 ± 0.13	0.71 ± 0.11
	(%BW)	0.36 ± 0.05	0.31 ± 0.02	0.32 ± 0.04	0.14 ± 0.03	0.16 ± 0.02	0.16 ± 0.04	0.33 ± 0.04	0.31 ± 0.01	0.41 ± 0.13
Lung	(g)	1.50 ± 0.27	1.32 ± 0.13	1.39 ± 0.18	1.42 ± 0.21	1.72 ± 0.95	1.60 ± 0.34	1.40 ± 0.17	1.28 ± 0.13	1.71 ± 0.64
	(%BW)	0.13 ± 0.01	0.13 ± 0.03	0.14 ± 0.02	0.32 ± 0.04	0.42 ± 0.27	0.37 ± 0.06	0.15 ± 0.02	0.15 ± 0.03	0.17 ± 0.02
Kidney	(g)	2.60 ± 0.24	2.71 ± 0.24	2.59 ± 0.15	2.67 ± 0.25	2.90 ± 0.28	2.85 ± 0.46	2.83 ± 0.18	2.85 ± 0.28	2.87 ± 0.36
	(%BW)	0.63 ± 0.05	0.64 ± 0.05	0.59 ± 0.04	0.61 ± 0.04	0.70 ± 0.08	0.65 ± 0.07	0.67 ± 0.02	0.68 ± 0.07	0.70 ± 0.04
Stomach	(g)	1.69 ± 0.36	1.63 ± 0.38	1.79 ± 0.30	1.88 ± 0.26	1.89 ± 0.15	1.81 ± 0.23	2.00 ± 0.44	2.18 ± 0.23	2.02 ± 0.19
	(%BW)	0.40 ± 0.06	0.38 ± 0.08	0.41 ± 0.07	0.43 ± 0.05	0.46 ± 0.04	0.42 ± 0.04	0.48 ± 0.11	0.52 ± 0.04	0.50 ± 0.03
Brain	(g)	1.75 ± 0.30	1.89 ± 0.13	1.85 ± 0.18	1.90 ± 0.12	1.70 ± 0.51	1.72 ± 0.38	1.94 ± 0.10	1.80 ± 0.24	1.83 ± 0.29
	(%BW)	0.42 ± 0.06	0.43 ± 0.04	0.43 ± 0.04	0.44 ± 0.03	0.41 ± 0.12	0.45 ± 0.02	0.41 ± 0.09	0.43 ± 0.06	0.45 ± 0.07
Adrenal gland	(g)	0.06 ± 0.02	0.07 ± 0.02	0.06 ± 0.02	0.06 ± 0.01	0.07 ± 0.02	0.05 ± 0.00	0.07 ± 0.01	0.06 ± 0.02	0.06 ± 0.02
	(%BW)	0.01 ± 0.00	0.02 ± 0.00	0.01 ± 0.00	0.01 ± 0.00	0.02 ± 0.01	0.01 ± 0.00	0.02 ± 0.00	0.01 ± 0.00	0.01 ± 0.01
Thymus	(g)	0.18 ± 0.10	0.26 ± 0.08	0.16 ± 0.08	0.14 ± 0.05	0.21 ± 0.12	0.26 ± 0.11	0.19 ± 0.07	0.19 ± 0.03	0.02 ± 0.08
	(%BW)	0.04 ± 0.02	0.06 ± 0.02	0.04 ± 0.02	0.03 ± 0.01	0.05 ± 0.03	0.06 ± 0.02	0.05 ± 0.02	0.04 ± 0.01	0.04 ± 0.03
Genitals	(g)	3.32 ± 0.26	2.91 ± 0.30	3.15 ± 0.22	3.32 ± 0.36	3.07 ± 0.24	3.16 ± 0.16	3.18 ± 0.08	2.75 ± 1.13	2.98 ± 0.35
	(%BW)	0.80 ± 0.06	0.73 ± 0.09	0.68 ± 0.09	0.76 ± 0.09	0.73 ± 0.07	0.76 ± 0.04	0.73 ± 0.08	0.77 ± 0.03	0.73 ± 0.11

## Data Availability

This article has fully reflected all the data of this study.

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
