# Peer review of "Fibrous Roots of Cimicifuga Are at Risk of Hepatotoxicity"

_molecules, 2022, doi:10.3390/molecules27030938_

Round 1
Reviewer 1 Report
Strong points:
The experiments are well organized and performed. I know that this kind of experiments are laborious and need a lot of work to get the appropriate data. The paper covers various aspects of research in order to answer the question if the fibrous roots are in fact the toxic part of the material. They include except the in vivo experiments, biochemical and hematological analysis, UHPLC-Q-TOF-MS analysis of the active substances of Cimicifuga rhizome and fibrous roots, in vitro experiments and Western blot analysis.
Weak points:
Experiments include male rats but the drug material is going to be used only by women. It is not necessary to be included in the paper. Men do not use this material. So I think that the part that includes male rats is in redundance.
The rhizome extract as well as fibrous root extract induced the expression of NF-kB in female rats suggesting that the inflammation was due not only to fibrous roots but to rhizome extract as well. This suggests that long time exposure or increased dose may lead to increased risk of hepatotoxicity to female.
In Figure 3, arrows (arrowheads) should be added to point the lesions caused by fibrous roots extracts
Line 261: hungry change to starve
Author Response
Dear Editor and Reviewer:
Many thanks for your insightful comments and suggestions, this was a great help to our manuscript. We have made corresponding revision according to your advice. Words in red are the changes We have made in the manuscript.
The following is the answers and revisions we have made in response to the questions and suggestions on an item-by-item basis.
- Experiments include male rats but the drug material is going to be used only by women. It is not necessary to be included in the paper. Men do not use this material. So I think that the part that includes male rats is in redundance.
Answer: Cimicifuga has been used as a traditional herb for thousands of years, it is not only effective in relieving female climacteric syndrome, but also in anti-inflammatory, antibacterial, sedative and analgesic, and has recently been found to have significant anti-cancer effects. When used for these purposes, it's possible for men to use them as well. Therefore, we believe it is necessary to comprehensively investigate its safety in both males and females.
- The rhizome extract as well as fibrous root extract induced the expression of NF-kB in female rats suggesting that the inflammation was due not only to fibrous roots but to rhizome extract as well. This suggests that long time exposure or increased dose may lead to increased risk of hepatotoxicity to female.
Answer: We have carefully considered your opinion, and this is indeed an important finding of our experimental results, which we described in line 244-248.
- In Figure 3, arrows (arrowheads) should be added to point the lesions caused by fibrous roots extracts
Answer: We re-uploaded Figure 3 and added arrows to the hepatocellular lesions in the fibrous root group.
- Line 261: hungry change to starve
Answer: We’ve changed the “hungry” to “starve”.
Reviewer 2 Report
Review on “Fibrous roots of Cimicifuga are at risk of hepatotoxicity”
This paper describes a comparison test on hepatotoxicity of rhizome and fibrous roots of Cimicifuga dahurica. In this manuscript, chemical compound identification using UHPLC-Q/TOF-MS, in vitro cytotoxicity test using L-02 cells, and in vivo toxicity test using rats were employed. The research subject of this manuscript is interesting, and this paper would be helpful to readers of Molecules. Therefore, reviewer thinks this paper would be acceptable for publication on Molecules. If following minor defects are revised appropriately, this paper would be more improved.
- In figure 1 and Tables 1 and 2, the authors identified many peaks on chromatograms detected using UHPLC-Q/TOF-MS. Nevertheless, there are many abundant unknown peaks on chromatograms of rhizome and fibrous roots of C. dahurica. If possible, several abundant unknown peaks should be identified.
- The authors should check abbreviations, according to their orders in this manuscript. Full names should be provided before use of abbreviations.
- Following mistyping should be checked.
On line 15, RFC → FRC
On line 43, ; → .
On line 47, we → We
On line 81, UPLC → UHPLC
On line 189, period (.), semicolon (;)
On line 247, ultra-performance → ultra-high-performance
On line 257, CO2 → CO2 (subscript)
On line 258, period (.)
Author Response
Dear Editor and Reviewer:
Many thanks for your insightful comments and suggestions, this was a great help to our manuscript. We have made corresponding revision according to your advice. Words in red are the changes We have made in the manuscript.
The following is the answers and revisions we have made in response to the questions and suggestions on an item-by-item basis.
- In figure 1 and Tables 1 and 2, the authors identified many peaks on chromatograms detected using UHPLC-Q/TOF-MS. Nevertheless, there are many abundant unknown peaks on chromatograms of rhizome and fibrous roots of C. dahurica. If possible, several abundant unknown peaks should be identified.
Answer: Seriously considered your suggestion, according to the Scifinder database of cimicifuga species and our laboratory's previous experience in the isolation and identification of cimicifuga, we summarized the fragmentation regularities of these compounds by mass spectrometry. And the mass spectrometry data of some reference standards were combined. Finally, we identified and supplemented the abundant unknown peaks in figure 1 and Tables 1 and 2, 52 compounds in rhizome and fibrous roots of C. dahurica were identified respectively.
- The authors should check abbreviations, according to their orders in this manuscript. Full names should be provided before use of abbreviations.
Answer: We carefully check all abbreviations in the manuscript and provide full names when they first appear.
- Following mistyping should be checked.
On line 15, RFC → FRC
On line 43, ; → .
On line 47, we → We
On line 81, UPLC → UHPLC
On line 189, period (.), semicolon (;)
On line 247, ultra-performance → ultra-high-performance
On line 257, CO2 → CO2 (subscript)
Answer: We have checked and corrected this mistyping you mentioned in our manuscript.